# Ballistic Properties and Izod Impact Resistance of Novel Epoxy Composites Reinforced with Caranan Fiber (*Mauritiella armata*)

**DOI:** 10.3390/polym14163348

**Published:** 2022-08-17

**Authors:** Andressa Teixeira Souza, Lucas de Mendonça Neuba, Raí Felipe Pereira Junio, Magno Torres Carvalho, Verônica Scarpini Candido, André Ben-Hur da Silva Figueiredo, Sergio Neves Monteiro, Lucio Fabio Cassiano Nascimento, Alisson Clay Rios da Silva

**Affiliations:** 1Department of Materials Science, Military Institute of Engineering, Rio de Janeiro 22290-270, RJ, Brazil; 2Engineering of Natural Resources of the Amazon Program, Federal University of Para, Ananindeua 67030-007, PA, Brazil

**Keywords:** caranan fiber, epoxy composite, ballistic armor, Izod impact test

## Abstract

Natural lignocellulosic fibers (NFLs) possess several economic, technical, environmental and social advantages, making them an ideal alternative to synthetic fibers in composite materials. Caranan fiber is an NFL extract from the leafstalk of the *Mauritiella armata* palm tree, endemic to South America. The present work investigates the addition of 10, 20 and 30 vol% caranan fiber in epoxy resin, regarding the properties associated with Izod notch tough and ballistic performance. Following ASTM D256 standards, ten impact specimens for each fiber reinforcement condition (vol%) were investigated. For the ballistic test, a composite plate with 30 vol%, which has the best result, was tested with ten shots, using 0.22 ammunition to verify the energy absorption. The results showed that when compared to the average values obtained for the epoxy resin, the effect of incorporating 30 vol% caranan fibers as reinforcement in composites was evident in the Izod impact test, producing an increase of around 640% in absorption energy. Absorbed ballistic energy and velocity limit results provided values similar to those already reported in the literature: around 56 J and 186 J, respectively. All results obtained were ANOVA statistically analyzed based on a confidence level of 95%. Tukey’s test revealed, as expected, that the best performance among the studied impact resistance was 30 vol%, reaching the highest values of energy absorption. For ballistic performance, the Weibull analysis showed a high R^2^ correlation value above 0.9, confirming the reliability of the tested samples. These results illustrate the possibilities of caranan fiber to be used as a reinforcement for epoxy composites and its promising application in ballistic armor.

## 1. Introduction

Selecting materials that industries can apply as a sustainable product is an important issue in modern engineering design [1]. A smart choice is natural polymer composites that provide productivity, ease of processing and lower costs [2,3]. By varying the different reinforcements and phases of the matrix, it is possible to obtain composite materials with varied properties and unique qualities [4,5].

Natural lignocellulosic fibers (NLFs), when used as a reinforcement in polymer composites, generate products that can be used in various industrial applications, such as: automotive, packaging, maritime, construction and military equipment [6,7]. NLFs attractive features are its low weight, biodegradability, renewability, societal benefits, non-toxic material, lower cost and good mechanical properties [8,9]. However, its impairing factors must be taken into account. These include high dispersion of physical properties, hydrophilicity and inhomogeneity inherent to the plant fiber structure [10]. The same fiber species, for example, depends on factors such as origin, age, fiber diameter and preconditioning, which might considerably affect their properties [11].

Research has shown that polymer composites reinforced with NLFs have ballistic efficiency comparable to synthetic aramid fabrics such as Kevlar™ [10,12,13,14]. Additionally, the abundance of NLF provides a cost-effective solution to the growing demand in the global protective armor sector. These sustainable alternatives can contribute to armor technology by making it more affordable and, thus, protect human life.

Today, there is an increased interest in many industrial sectors for the commercial use of NLF-based composites. In particular, lesser known NLFs such as guaruman [15], copernicia prunifera [16], cyperus malaccensis [17], and fique [18] have been recently studied as reinforcements of polymer composites. Another less known NLF is the caranan fiber extract from the leafstalk of the *Mauritiella armata* palm tree, endemic to South America. As far as we know, only a single research paper on caranan fiber incorporated polymer composites has been reported by Souza et al. [19]. The paper presents results of 10, 20 and 30 vol% epoxy-caranan composites in dynamic mechanical analyses (DMA), thermal studies (DSC/TGA) and tensile tests. According to the authors, the modulus of elasticity (50%), total elongation (40%) and tensile strength (130%) of the composites increased with 30 vol% of caranan fibers incorporated into the matrix. The maximum working temperature of the composite and the fiber was found to be 200 °C. DMA results showed 62–65 °C to be at the lower limits, and 96–113 °C upper limits for Tg.

Based on the results previously studied, which present interesting specific properties and similarities to other NLFs already studied in the literature, it was decided that a more in-depth observation should be carried out on this lesser-known caranan fiber. Therefore, in order to continue the results already obtained, in the present work, the possible use of caranan fiber as reinforcement for epoxy composites is investigated for the first time regarding its impact resistance during Izod tests, as well as its potential in ballistic tests. However, the lack of information in the literature about caranan fiber makes it necessary to conduct a more comprehensive study by means of flexural tests and its ballistic performance as a component of a multilayered armor system (MAS).

## 2. Materials and Methods

### 2.1. Materials

The caranan fibers used in this work were donated by the Federal University of Pará (UFPA). After manual extraction with a sharp razor, the leaf petioles, Figure 1a, were cleaned and defibrillated, Figure 1b. The average diameter found for caranan fibers was 0.79 µm. The isolated fibers were cut to a length of 150 mm and oven dried at 70° C for 24 h or until their weight remained stable, Figure 1c.

The polymer used to produce the composite matrix was an epoxy resin diglycidyl ether of the bisphenol A (DGEBA), the hardener applied to the resin was triethylene tetramine (TETA) with a stoichiometric ratio of 100 parts of epoxy to 13 parts of TETA, both produced by Dow Chemical, São Paulo, and distributed by Epoxyfiber, Rio de Janeiro, Brazil.

### 2.2. Composites Processing

To produce the composite plates, with proportions of 10, 20 and 30% by volume of caranan fiber, a metal mold with rectangular dimensions of 150 × 120 × 12 mm^3^ were used. The data presented in the literature provided DGEBA/TETA density of approximately 1.1 g/cm^3^ [20]. For caranan fiber density, previous studies indicate a density of 0.66 g/cm^3^ [19].

The fibers were carefully layered in the mold and a previously calculated mixture of resin and hardener was added to fill the voids. The final plate was obtained by processing in a SKAY hydraulic press (Skay Industry, São Paulo, Brazil) under a load of 5 tons for at least 24 h.

### 2.3. Izod Impact

The Izod impact test was performed according to ASTM-D256 [21]. The objective was to measure the fracture energy of the composites in Joules per meter (J/m). Ten (10) specimens were prepared with dimensions of 63.5 × 12.7 × 10 mm^3^, as shown in Figure 2b, 45 ± 1° and 2.54 mm-deep, as shown in Figure 2a. The equipment used for the test was a Pantec (Rio de Janeiro, BR, Brazil) model XC.50 with an 11 J hammer from the State University of Norte Fluminense (UENF).

### 2.4. Ballistic Tests

For the ballistic tests, a gunpower SSS (Condor, Ashford, UK) pressure rifle with a standard weapons noise suppressor was used. The projectile is a 22-gauge lead with an estimated mass of 3.3 g. To determine the absorption energy, an air chrony model MK3 ballistic chronograph (Air Chrony, Nové Město, Czech Republic) with an accuracy of 0.15 m/s was used to measure the impact velocity, and a ProChrono model Pal ballistic chronograph (Competition Eletronics, Rockford, IL, USA) with an accuracy of 0.31 m/s was used to measure the residual velocity.

The air rifle was positioned 5 m away from the target, consisting of a plate held in a vise and aligned perpendicularly to the rifle. One ballistic chronograph was positioned 10 cm before the target and the other was placed 10 cm behind the target. The energy absorbed by the target was calculated using Equation (1) [22].
(1)$$E_{abs}=\frac{m\left(V_{i}^{2}-V_{r}^{2}\right)}{2}-E_{abs}^{*}$$
where *m* is the projectile mass, $V_{i}$ is the impact velocity, $V_{r}$ residual velocity and $E_{abs}^{*}$ is the energy absorbed without the sample. Another important parameter obtained from this test is the limit velocity $V_{L}$ which is the root of the ratio between absorbed energy and mass, Equation (2) [23].
(2)$$V_{L}=\sqrt{\frac{2E_{abs}}{m}}$$

### 2.5. Micrography Analysis

Scanning electron microscopy (SEM) images of Izod impact-ruptured composite specimens and ballistic tests were analyzed. For this, a Quanta FEG 250 model FEI microscope (Field electron and Ion Co., Hillsboro, OR, USA) was used. The samples were fixed on a carbon ribbon and later sputtered with gold. The working distance used was 15.5 mm and secondary electrons were accelerated with 25 kV.

### 2.6. Statistical Validation

The results obtained in this work were treated to ensure reliability within a limit of 95% confidence. For this, analysis of variance (ANOVA), Tukey test and Weibull analysis were used.

### 2.7. Weibull Analysis

The Weibull distribution is based on the principle of maximum likelihood and is the most common statistical function in engineering applications that requires a level of reliability. Its distribution can be defined by the density function given by Equation (3) [24].
(3)$$F\left(x\right)=1-exp{\left(\frac{x}{\theta }\right)}^{\beta }$$
where θ is the Weibull characteristic unit or scale parameter and β the Weibull modulus or shape parameter.

### 2.8. Analysis of Variance (ANOVA)

All ANOVA results were obtained from the Minitab^®^ Statistical Software application (Minitab, LLC., State College, PA, USA). The most used mean comparison test, Tukey, was chosen because it has a significance level of α = 5% (α = 0.05), as follows [25].
(4)$$HSD=q\sqrt{\frac{EMS}{r}}$$
where *q* is the constant tabulated, *EMS* = error mean square of the ANOVA and *r* the number of repetitions of each treatment.

## 3. Results and Discussion

### 3.1. Izod Impact

Table 1 shows all the results obtained for the Izod impact test of the different composites materials and neat epoxy resin.

The values found are similar to those already reported in the literature for other NLFs composites [26,27]. The specimens of neat epoxy resin, Figure 3a, and composites with 10 vol% fiber, Figure 3b, were all completely fractured and, thus, were validated by the standard.

Specimens of 20 and 30 vol% fiber, Figure 3b,c, respectively, showed incomplete fracture. However, the fibers were broken and pulled out. Although the ASTM D256 [21] standard invalidates the test in the absence of total fracture, there are indications that the energy supported by the composite was equal to or greater than that recorded by the pendulum. In other words, as there was no complete rupture, the absorbed energy was not enough to fracture the composite.

The results in Table 1 can be better visualized in the graph in Figure 4. With the increase in the fiber fraction, it is possible to notice an improvement in the impact energy, as already observed for other NFLs [27,28]. The high dispersion of the values given by the error bars associated with the fiber is also a known characteristic of NLFs, since they have a great heterogeneity [18,29].

When compared to the average values obtained for the neat epoxy resin, the effect of incorporating 30% by volume of caranan fibers as reinforcement in composites was evident, producing an increase of 637.30%.

A statistical analysis of ANOVA, Table 2, was performed to compare the averages obtained and to verify if there was a significant difference in energy absorption between them.

Therefore, with a 95% confidence level (or 5% significance level), the hypothesis that the treatments have equal means is rejected, because the F_calc_ was much higher than the F_tab_ (critical).

Table 3 shows the results of Tukey’s test. The calculated HSD was 48.36 J/m, and, thus, the differences above the HSD were considered significant. These values are marked in bold and showed that the impact strength of the 30 vol% caranan fiber composites was better than all other tested specimens.

To better understand the behavior of the composites analyzed in ballistic impact, Figure 5 presents SEM images of broken epoxy-caranan (30 vol%) specimens.

In these images, different fracture mechanisms can be observed. One might note, Figure 5, that cracks (resin failure) are formed in the matrix and their paths are blocked and interrupted by the fibers. Additionally, in this SEM, good fiber adhesion can be seen at the fiber-matrix interface. Similar behavior was reported by Costa et al. [30] and Junio et al. [16]. Other failure mechanisms that are influenced by fibers are also associated with the fragility of the polymer matrix failure mechanism. These fracture mechanisms are more complex and include the rupture of caranan fibers. Their pullout is evidenced by the circular holes shown in the fractography, Figure 5b. These results can be related to the absorption of high impact energies, shown in Table 1 and Figure 4.

Figure 6 shows a comparison between good and poor fiber adhesion to the resin. To the left of the fiber, when the fiber-matrix interface is not adequate, the matrix rupture can be observed. This failure mechanism is not observed on the right side of the fiber. The difference in the nature of the polymer matrix and natural fiber explains this behavior. While caranan fibers, such as other NLFs, exhibit a hydrophilic nature, the epoxy resin has a hydrophobic character. This difference in nature impairs the interfacial adhesion of the reinforcement in the matrix, facilitating the delamination mechanism. Additionally, river marks are stopped when approaching the fiber. The sum of all these mechanisms directly contributes to the increase in energy absorption of the composite [13,31], showing the reinforcement of the caranan fiber in the epoxy matrix.

Due to the heterogeneity of natural materials, the same fiber might have different characteristics. This can occur for several reasons, among them we can highlight plant cultivation, plant age, fiber roughness, surface flaws, diameter variation, climate, extraction procedures and soil. Such factors may have influenced the results related to the fiber/matrix interface [32].

Pullout testing was performed to characterize the caranan fiber/epoxy bonding and revealed a rather high interfacial shear strength of 17 MPa. As a result, one should anticipate good fiber/matrix adhesion and the possibility of caranan fiber reinforcing behavior [19].

### 3.2. Ballistic Tests

In order to estimate the ballistic behavior of the 30 vol% composite of caranan fiber, a residual velocities test was performed. Table 4 presents the absorption energy values for each one of the ballistic shots performed on the plate. During the tests, all the specimens were perforated so their residual velocities could be measured.

When considering the post-impact aspect of the plate, i.e., the physical integrity of the composite after the ten shots, Figure 7, the sample did not fracture. Indeed, a very important criterion for application in ballistic protection [33,34] is the ability of the plate to receive projectile impact without disintegration.

Table 5 presents the average values of the composite mass *m_c_*, projectile mass *m_p_*, average impact velocity *V_i_*, average residual velocity *V_r_*) and absorption energy *E_abs_* of each composition (C30%—composite reinforced with 30 vol% of fiber).

Based on the results obtained, it can be inferred that the neat epoxy resin presenting the highest energy of absorption failed to resist an expressive number of shots, quickly suffering a rupture [14]. For application in ballistic shielding, this is a negative aspect because target integrity is one of the evaluation criteria for effective protection. The higher *E_abs_* value observed for neat epoxy may be associated with its fragility, which tends to dissipate energy by generating fractured surfaces. This can be considered an indication that the fiber reinforcement was not carried out effectively or with adequate volume [35].

For comparison, some limited velocity results, found in the literature for shooting tests with a 0.22-inch caliber, are presented in Table 6. It is noteworthy that these results were found for shots with a compressed air apparatus and not with a firearm. The composites of the present work presented similar results of *V_L_* when compared to the other related NFL.

After the ballistic impact, the brittle behavior of the epoxy matrix can be verified in Figure 8. Such behavior is evidenced by the appearance of river marks and cracks in the epoxy matrix [28,36,37,38]. River marks on the surface usually means restricted plasticity on the crack tips and very quick crack propagation. However, no fiber failure is observed in this SEM, thus evidencing its reinforcement in the epoxy matrix by the caranan fiber.

Weibull statistical analysis, Table 7, was also performed. It is possible to verify that the points are within the adjusted line, justifying the high value of R^2^, above 0.9. In addition, it is worth noting that the characteristic value (θ) is similar to the average found for *E_abs_* of the composite.

Figure 9 presents the graph of this statistical analysis for better visualization.

The results of this Weibull statistical analysis, Figure 9, indicated the good reliability of the obtained results and revealed a homogeneous characteristic of the individual samples.

## 4. Conclusions

Compared to the neat epoxy sample, composites displayed a significant increase in impact energy when reinforced with caranan fibers. The best result was obtained incorporating 30 vol%, which produced an increase in Izod absorbed impact energy of about 640%. Fracture mechanisms observed in SEM images and in macroscopic specimens showed the predominant breakage of the brittle epoxy matrix. Other failure mechanisms observed were fiber breakage, pullout mechanism and delamination. The epoxy composite reinforced with 30 vol% of caranan fiber showed excellent results in terms of plate integrity after the residual velocity test. Nevertheless, the energy absorption, when compared to neat epoxy resin, showed inferior results. This is due to the resin’s failure mechanism and its fragile behavior. The limit speed of 186 m/s was similar to studies already carried out in the literature with polymer composites reinforced with other NFLs. Statistical analyses confirmed the reliability of the data, retained a safety limit of 95% to be applied to the studied samples.

## Figures and Tables

**Figure 1 polymers-14-03348-f001:**
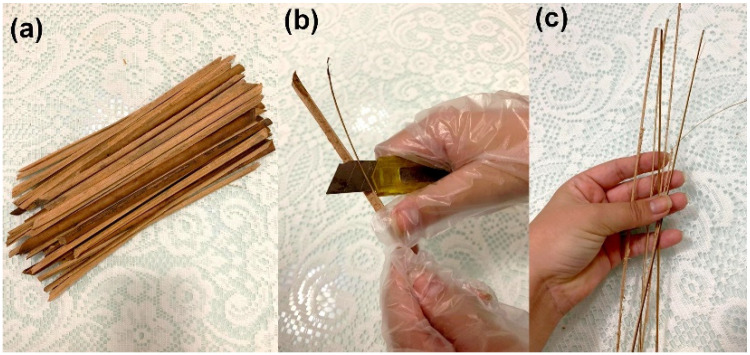
Processing of caranan fiber: (**a**) leafstalk, (**b**) mechanical separation of fibers, and (**c**) defibrillated caranan fibers.

**Figure 2 polymers-14-03348-f002:**
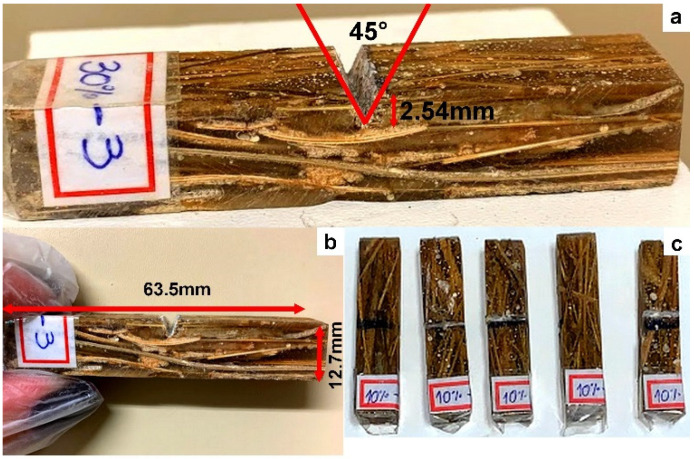
Izod impact test specimens. (**a**) measure of the notch (**b**) measure of length and thickness (**c**) some samples of 10% vol.

**Figure 3 polymers-14-03348-f003:**
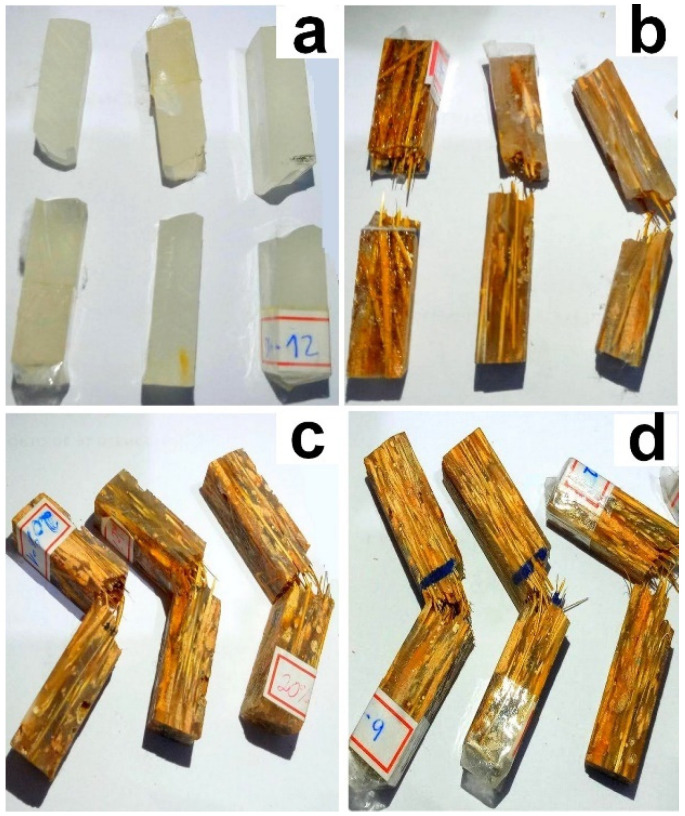
Izod specimens tested and categorized by caranan fiber percentage volume: (**a**) 0% (neat epoxy resin); (**b**) 10 vol.%; (**c**) 20 vol.%; (**d**) 30 vol.%.

**Figure 4 polymers-14-03348-f004:**
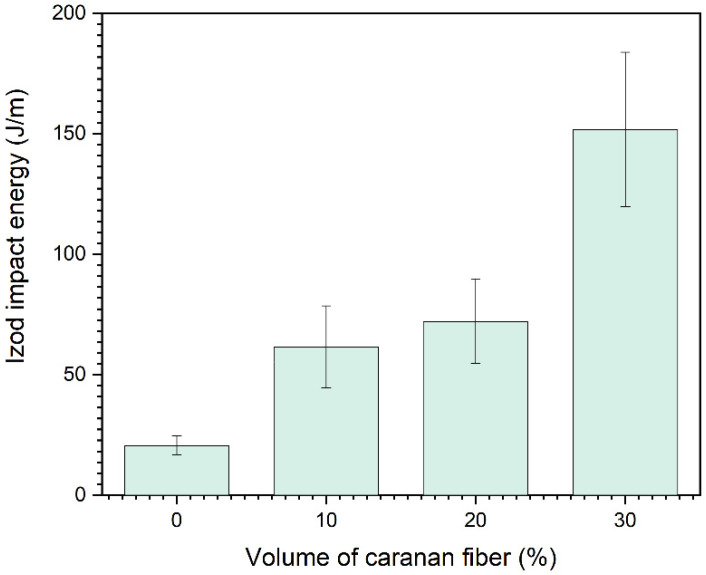
Izod impact energy as a function of the fiber fraction for the neat epoxy resin and caranan fiber-reinforced composites an increase of 637%.

**Figure 5 polymers-14-03348-f005:**
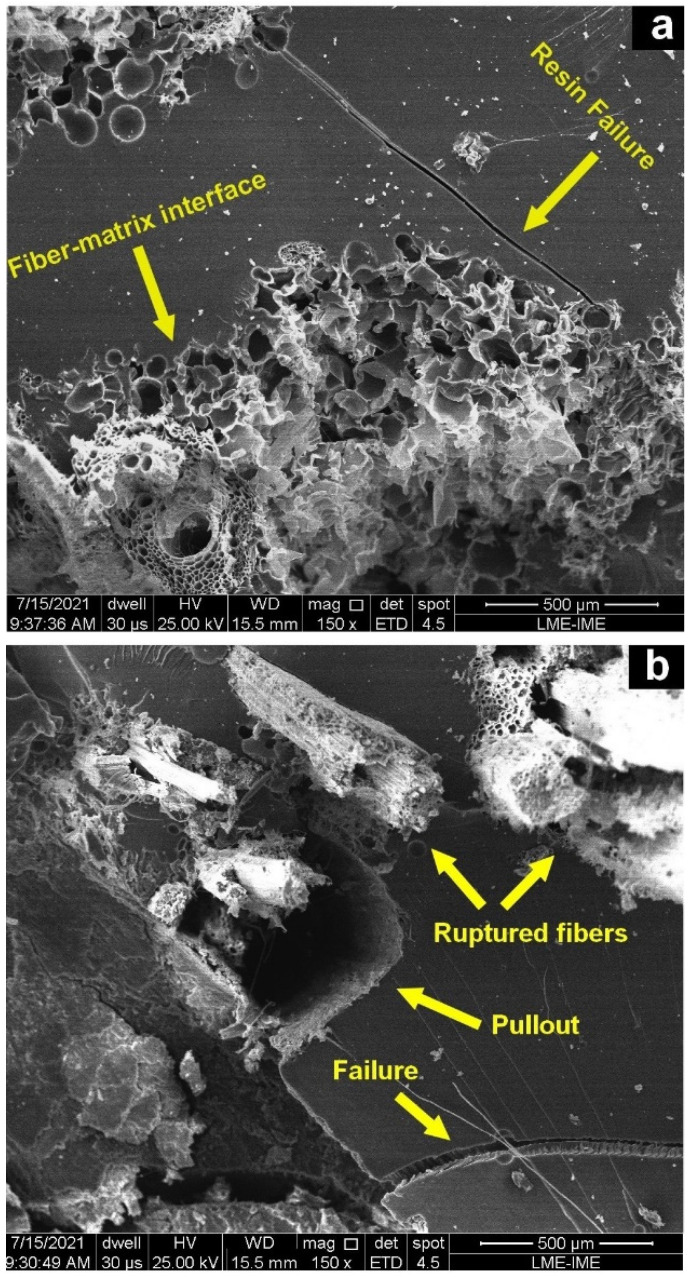
SEM images of composites reinforced with 30 vol% of caranan (**a**) fractured surfaces fibers (**b**) fiber pullout.

**Figure 6 polymers-14-03348-f006:**
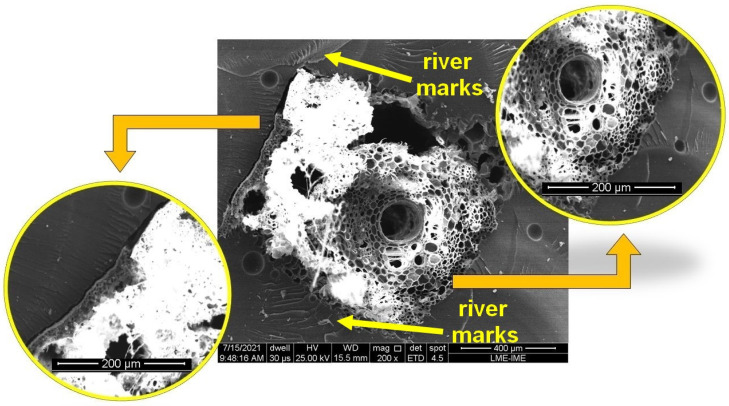
SEM images of the fiber-matrix interface of the 30 vol% epoxy-caranan composite after the Izod impact test.

**Figure 7 polymers-14-03348-f007:**
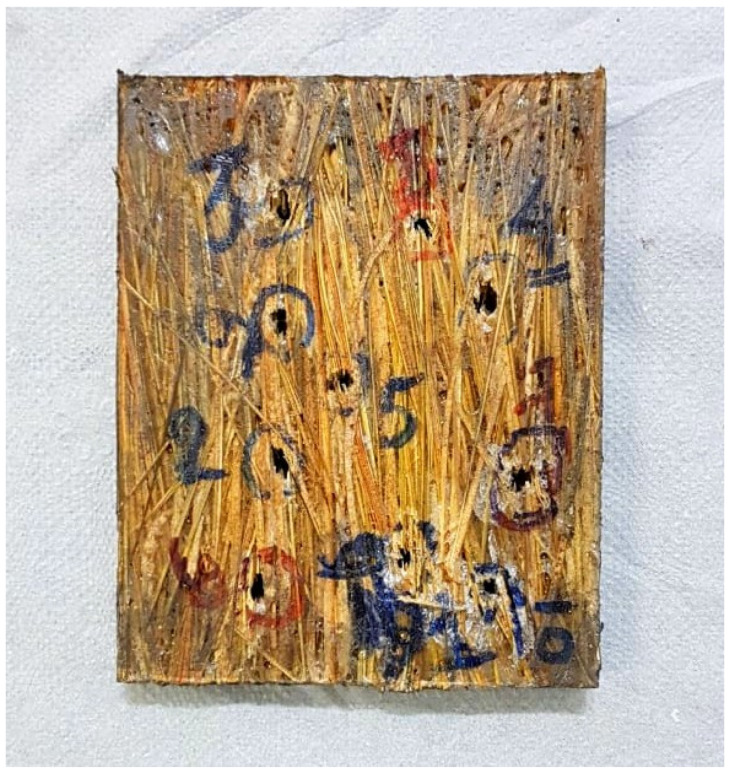
Target subjected to ballistic impact 30 vol% epoxy-caranan fibers composites.

**Figure 8 polymers-14-03348-f008:**
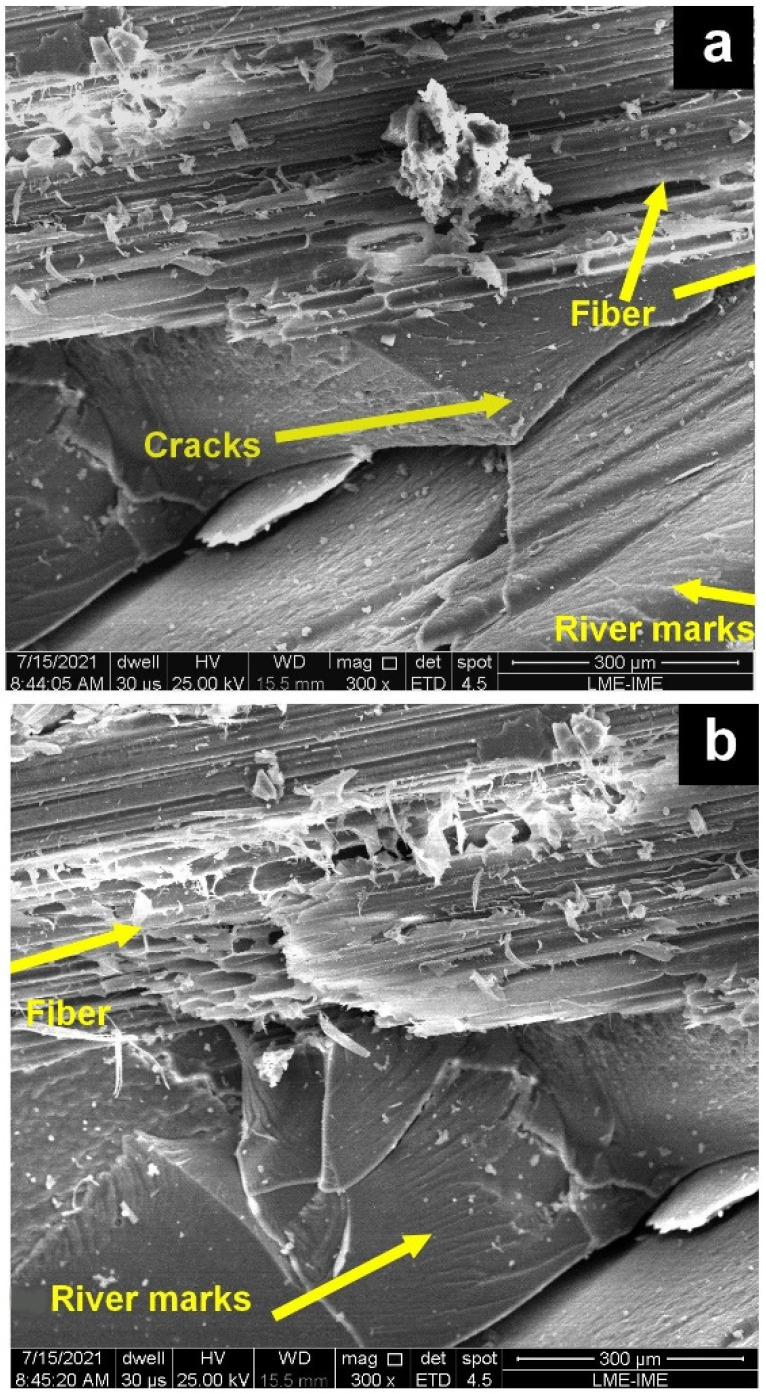
SEM images of 30 vol% epoxy-caranan composites after ballistic testing (**a**) fractured surfaces (**b**) river marks in epoxy matrix.

**Figure 9 polymers-14-03348-f009:**
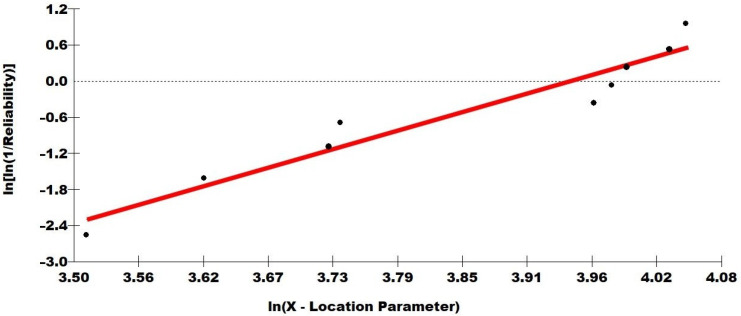
Weibull frequency distribution graph for the epoxy composites with 30 vol% of caranan fiber.

**Table 1 polymers-14-03348-t001:** Izod impact energy (J/m) for neat epoxy resin and caranan fiber reinforced composites.

Specimen	Neat Epoxy Resin (J/m)	10% (J/m)	20% (J/m)	30% (J/m)
1	16.05	71.63	53.47	100.28
2	17.45	76.56	59.43	163.44
3	16.72	66.89	71.63	126.30
4	20.90	69.15	107.44	162.03
5	23.14	46.26	100.28	138.88
6	23.14	92.59	61.72	114.63
7	16.72	37.14	77.16	214.89
8	18.24	46.29	61.72	185.73
9	26.17	41.49	57.30	154.32
10	27.37	68.25	71.63	157.59
Average	20.59 ± 3.95	61.61 ± 16.98	72.18 ± 17.35	151.81 ± 32.04

**Table 2 polymers-14-03348-t002:** Analysis of variance for the tensile strength of the caranan fiber-reinforced composites.

Variation	Degrees of Freedom	Sum of Squares	Mean Square	F_calc_	F_tab_
Treatments	3	90379.65	30126.55		
Residue	36	16315.47	5438.49	5.54	2.80
Total	39	106695.12			

**Table 3 polymers-14-03348-t003:** Tukey’s test for the Izod impact energy of the neat epoxy resin and caranan fiber-reinforced composites.

Vol% Caranan Fiber	0	10	20	30
0	-	41.02	51.59	131.22
10	41.01	-	10.57	90.24
20	51.56	10.57	-	79.63
30	131.22	90.20	79.63	-

**Table 4 polymers-14-03348-t004:** Absorption energy in ballistic testing of a 30 vol% fiber composite sample.

Performed Shoots	Absorbed Energy (J)
1	54.32
2	6.44
3	41.61
4	42.03
5	37.21
6	53.61
7	52.75
8	57.29
9	33.49
10	53.11
Average	48.17 ± 8.25

**Table 5 polymers-14-03348-t005:** Absorption energy in ballistic testing of a 30 vol% fiber composite sample.

Specimen	*m_p_* (g)	*V_i_* (m/s)	*V_r_* (m/s)	*E_abs_* (J)	Shoots
No specimen	3.29 ±0.07	287.25 ± 4.59	286.18 ± 4.00	1.08 ± 0.90	-
Neat epoxy	56.44 ± 0.05	290.55 ± 2.47	140.82 ± 34.05	106.81 ±16.69	3
C30%	3.34 ± 0.03	283.39 ± 4.28	226.56 ± 10.91	48.17 ± 8.25	10

**Table 6 polymers-14-03348-t006:** Limit velocity for 30 vol% epoxy-caranan and other NLF-reinforced composites in the literature.

Composite	*V_L_* (m/s)	Reference
Epoxy-caranan fibers 30%	186.00	Present Work
Epoxy-tucum fibers 20%	224.49	33
Epoxy-tucum fibers 40%	204.37	33
Epoxy-Cyperus Malaccensis fibers 30%	212.5	17
Epoxy-Cannabis Sativa hemp fabric 30%	256.3	34

**Table 7 polymers-14-03348-t007:** Weibull distribution for energy absorbed from the epoxy-caranan composite 30 vol%.

Composite	θ	*β*	R^2^
Epoxy-caranan fibers 30%	51.64	5.304	0.9263

## Data Availability

Not applicable.

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
