# Peer review of "Ballistic Properties and Izod Impact Resistance of Novel Epoxy Composites Reinforced with Caranan Fiber (Mauritiella armata)"

_polymers, 2022, doi:10.3390/polym14163348_

Round 1
Reviewer 1 Report
Overall, this paper present a good finding that is worth for publication, however, some improvement is necessary before this paper is ready to be published.
Title: The title should be revise by removing the “Promising” words since journal articles should report scientific findings without bias point of view.
Abstract:
- General introduction of the study should be provided prior to the objective of study.
- The information provided in the abstract is sufficient, however, the writing should be improved to make it sound more scientifically.
- Grammar and punctuation should be revise as well.
Introduction:
- The introduction is too general on the use of NLF, author should provide a specific paragraph to introduce the interest of this study i.e
o the application of NLF in ballistic application
o Introduction on caranan and other study related to this plant
- At the end of introduction, author should highlight the gap of study and the interest which trigger this research to be carried out.
- Grammar and punctuation should be revise accordingly.
-
Materials and methodology:
- Please describe the average diameter of the fiber.
- Please state the name and country for all the machine use in this study.
- Grammar and punctuation should be revise accordingly.
Results and Discussion
- Figure 3 is not appropriate, please provide appropriate labelling, border, and uniform Figure size.
- Table 2 suddenly show data of treatment and residue, where does this come from?
- Author states that Figure 6 shows comparison of good and poor fiber adhesion, however, there is no different treatment conducted on the fiber surface, how does a same fiber shows two different properties in the same matrix? This phenomenon might occur after impact and should not be associated with difference in the nature of polymer matrix and natural fiber, since it is the same fiber and same matrix in the same picture. Author have to relook at this discussion since it is not accurate and might be misleading to the reader.
- Nevertheless, author have provide a good discussion on the reason for improvement, i.e river marks are stopped when approaching the fiber. However, the labelling on this river mark is missing, please add on the Figure.
- In addition, author can relate the improvement in the strength of the fiber due to fiber pullout mechanism.
- Table 6 were presented, however, there is no discussion provided and the intention of showing this table is not expressed well.
- In Figure 8, author should focus on the role of fiber in the composites rather than expressing the brittle behavior of the epoxy.
- Grammar and punctuation should be revise accordingly.
-
-
Conclusion
- Please revise this sentence as it is quite difficult to be understood “Fracture mechanisms observed in SEM images and in macrosopic specimens showed the predominant rupture of the brittle epoxy matrix, changing to fiber break, pullout mechanism and delamination” what does author means by changing to fiber break etc…?
- Second point: FNL or NLF
- The ballistic test show negative quantitative results compare to the epoxy, author have to highlight this as well.
- How does statistic analysis for ballistic show caranan fiber improve the properties? Since there is no quantitative data for the improvement other than good integrity of the sample.
- Grammar and punctuation should be revise accordingly, and author might consider to write only one paragraph for the conclusion.
-
Author Response
The response is attached.

Reviewer 2 Report
In this research caranna-epoxy composites reinforced with 10, 20 and 30 vol% were investigated for mechanical properties associated with Izod notch toughness and ballistic performance. the effect of incorporating 30vol% caranna fibers as reinforcement in composites was evident in the Izod impact test, producing an increase around of 640% in absorption energy . the predominant rupture of the brittle epoxy matrix, changing to fiber break, pullout mechanism and delamination. with a confidence level of 95% (significance level 0.5%), that the incorporation of caranna fibers increasingly reinforces the epoxy matrix. caranna fiber as a reinforcement for epoxy composites and its possible applicability in ballistic protection. However, I considered it can be published in Polymers after a minor revision:
1. I think the content of the introduction can be enriched, such as what is the research content of caranna fiber incorporated polymer composites, and what in-depth research has been done on this basis.
2. The article says that the quality is stable after drying, which means that the quality changes during the drying process. I think it is possible to add a before-and-after comparison chart of the quality to further illustrate the quality problem.
3. In the part of composite material processing, I think the specific steps of preparing composite material board can be more detailed, and the ratio of resin and hardener should be clearly stated. Only some samples of 10% vol are shown in Figure 2, which does not play a role in volume control. In Figure 2c, some samples have gaps, and the other part has no gaps. I think it should be explained what causes this.
4. I think equations 1-4 should be clearly referenced. In the Izod impact test, all samples fractured, and the impact energy increased with increasing volume, I think 50 vol% of caranna fibers can be set for further comparison.
5. The integrity of the plate is one of the evaluation criteria for effective protection, but after the sample is impacted, it can be observed that the sample is penetrated, whether this meets the requirements of bulletproof materials, or how to improve this situation, I think the author should be more explain in detail.
6. The language of the article still needs to be polished, and grammatical errors need to be corrected. add some references such as “Journal of Cleaner Production, 359 (2022) 132134., Journal of Materials Science & Technology, 2022, 127:153-163.,”
Author Response
The response is attached

Reviewer 3 Report
The authors tried to reinforce epoxy with caranan fiber from Mauritiella armata. After screening the manuscript, I think it can be considered for publication in Polymers. However, the authors should more clearly describe the novel and the significance of their study. Here are some specific recommendations and questions for authors:
1. You claimed that there was a similar paper that also investigated the epoxy-caranan composites. The similarities and differences between your work and that of similar research should be more clearly presented in the Introduction section.
2. What are the similarities and differences between caranan fibers and other NLFs?

Author Response
The response is attached
